# Ionic Liquid-Modulated Synthesis of Porous Worm-Like Gold with Strong SERS Response and Superior Catalytic Activities

**DOI:** 10.3390/nano9121772

**Published:** 2019-12-12

**Authors:** Kaisheng Yao, Nan Wang, Zhiyong Li, Weiwei Lu, Jianji Wang

**Affiliations:** 1School of Chemical Engineering and Pharmaceutics, Henan University of Science and Technology, Luoyang 471023, China; 18848962198@163.com (N.W.); luwei1980@126.com (W.L.); 2Collaborative Innovation Center of Henan Province for Green Manufacturing of Fine Chemicals, Key Laboratory of Green Chemical Media and Reactions, Ministry of Education, School of Chemistry and Chemical Engineering, Henan Normal University, Xinxiang 453007, China; lizhiyong03@126.com

**Keywords:** ionic liquids, porous worm-like Au, SERS enhancement, nitroaromatic reduction

## Abstract

Porous gold with well-defined shape and size have aroused extensive research enthusiasm due to their prominent properties in various applications. However, it is still a great challenge to explore a simple, green, and low-cost route to fabricate porous gold with a “clean” surface. In this work, porous worm-like Au has been easily synthesized in a one-step procedure from aqueous solution at room temperature under the action of ionic liquid tetrapropylammonium glycine ([N_3333_][Gly]). It is shown that the as-prepared porous worm-like Au has the length from 0.3 to 0.6 μm and the width of approximately 100–150 nm, and it is composed of lots of small nanoparticles about 6–12 nm in diameter. With rhodamine 6G (R6G) as a probe molecule, porous worm-like Au displays remarkable surface enhanced Raman scattering (SERS) sensitivity (detection limit is lower than 10^−13^ M), and extremely high reproducibility (average relative standard deviations is less than 2%). At the same time, owing to significantly high specific surface area, various pore sizes and plenty of crystal defects, porous worm-like Au also exhibits excellent catalytic performance in the reduction of nitroaromatics, such as *p*-nitrophenol and *p*-nitroaniline, which can be completely converted within only 100 s and 150 s, respectively. It is expected that the as-prepared porous worm-like Au with porous and self-supported structures will also present the encouraging advances in electrocatalysis, sensing, and many others.

## 1. Introduction

Due to their various pore structures, large specific surface area, many transfer channels, and large number of active and accessible sites [1,2], porous Au have been paid increasing attention in recent years and found potentials in diverse areas including catalysis [3,4], sensing [5] and surface enhanced Raman scattering (SERS) [6,7]. Up to now, several strategies have been developed for the successful preparation of porous Au. Among these strategies, the dealloying of bulk Au-based alloys is the most commonly used method. For example, with AgAu alloy as a substrate, porous netlike Au has been constructed via the selective dissolution of Ag in a nitric acid solution [2,8,9,10,11]. At the same time, other methods were also explored for the preparation of porous Au. For instance, through laser ablation and electrophoretic deposition, netlike porous Au films have been prepared [12]; by using a hydrothermal method, a porous Au network structure could be constructed [13]; by anodization of Au in aqueous KCl solution [14] or oxalic acid solution [15], porous Au has been easily obtained; by the template method, a bowl-shaped porous Au structure was also reported [16]. Very recently, Shimasaki et al. [17] exhibited that mesoporous Au had been prepared through a seed-mediated method using mesoporous silica KIT-6 as a hard template. Nevertheless, in the strategies mentioned above, strong corrosive reagent (hazardous), expensive equipment, higher temperature, hard template and/or tedious experimental procedures have been always used. In addition, in the approaches of the dealloying of bulk Au-based alloy and anodization of bulk Au, the loss of precious metals is unavoidable, which inevitably leads to high costs, environmental problems and time consuming. Thus, it is critical to conceive a simple, mild, and low-cost route to grow and assemble well-defined porous Au nanostructures for the advanced applications.

Ionic liquids (ILs) have received great attention due to their low melting point, high ionic conductivity, high chemical and thermal stability, and extensive solubility [18]. Furthermore, the unique structure and ionic interaction in ILs endow them with unique physical and chemical properties, which have been widely applied in organic synthesis [19], extraction [20] and catalysis [21]. In inorganic synthesis, various nanomaterials, presenting well-defined shapes and sizes and superior performance, have been successfully prepared, in which ILs play irreplaceable roles [22,23,24,25,26,27,28,29]. For example, Feng et al. [30] synthesized AuPt nanodendrites under the stabilization of 1-aminopropyl-3-methylimidazole bromide ([APmim]Br). He et al. [31] fabricated a Pt cube by using 1-butyl-3-methylimidazole tetrafluoroborate ([C_4_mim][BF_4_]) to reduce the reduction rate of Pt(acac)_2_ and control the formation of the cube morphology. Recently, we obtained various Au nanostructures assisted by several functionalized ILs. It was found that not only ILs but also the functional groups in ILs had significant effects on product morphology and SERS responses [32]. These results encourage us to further explore new routes involving ILs for the fabrication of nanomaterials with well-defined structures and excellent performance.

Herein, a green and simple one-step route was employed for the growth and assembly of porous worm-like Au assisted by tetrapropylammonium glycine ([N_3333_][Gly]) in aqueous solution at room temperature. It was shown that [N_3333_][Gly] played important roles in constructing both porous nanostructures and worm-like morphology. To the best of our knowledge, it is the first time that tetraalkylammonium amino acid-based ILs was utilized to assist the growth of the special Au nanostructures and their growth process was proposed. Moreover, the as-prepared porous worm-like Au demonstrated outstanding SERS response and excellent catalytic performance for the reduction of nitroaromatics in water.

## 2. Materials and Methods

### 2.1. Materials

Tetramethylammonium hydroxide ([N_1111_][OH], 25%), tetrapropylammonium hydroxide ([N_3333_][OH], 25%) and tetrabutylammonium hydroxide ([N_4444_][OH], 25%) were purchased from Shanghai Chengjie Chem. Co. (Shanghai, China). Glycine (99%) was supplied by Tianjin Kermel Chem. Reagent Co. (Tianjin, China) Acetonitrile (≥99%) was obtained from Xilong Science. Co. Ltd. (Guangzhou, China) Ascorbic acid (≥99%) was purchased from Alfa Aesar Co. Sinopharm Chem. Reagent Co. (Beijing, China) provided chloroauric acid hydrated (HAuCl_4_·4H_2_O), ethanol and sodium borohydride (NaBH_4_, 98%). Aladdin Chem. Co. (Shanghai, China) provided *p*-Nitrophenol (*p*-NP, 98%), Rhodamine 6G (R6G, 95%), *p*-aminothiophenol (PATP, 97%), and pyridine (99.5%). Tianjin Damao Chem. Reagent Co. (Tianjin, China) sent *p*-Nitroaniline (*p*-NA, ≥99.5%) and methanol.

### 2.2. Methods

For a typical synthesis of ionic liquid, [N_3333_][OH] was added into an aqueous solution containing a slightly excess glycine under stirring at room temperature. Then, the mixed solution was rotatory evaporated at 60 °C to remove water. After cooling to room temperature, an acetonitrile/methanol mixed solvent with a volume ratio of 7:3 was added and stirred. The excessive glycine was gradually crystallised and then separated from the solvent through filtration. This procedure was repeated until glycine were no longer precipitated. After that, the solvent was distilled off at 60 °C on a rotary evaporator, and [N_3333_][Gly] was obtained through drying at 60 °C for 48 h under vacuum in the presence of P_2_O_5_. Similarly, tetrabutylammonium glycine ([N_4444_][Gly]) and tetramethylammonium glycine ([N_1111_][Gly]) were also prepared.

For the typical synthesis of porous worm-like Au, in 5.00 mL of water, a certain amount of [N_3333_][Gly] was added to form an IL aqueous solution (100 mM). Then, HAuCl_4_ was put into to create a 1 mM concentration in IL aqueous solution. After mixing thoroughly and letting stand for 10 min, 0.1 mL of 100 mM ascorbic acid in water was quickly put into the mixed solution at room temperature. After slightly shaking, the solution was left undisturbed for 10 h. Then, the porous worm-like Au was obtained by centrifugal separation and then washed with ethanol.

### 2.3. Characterization

Morphology of Au products was observed on a SU8010 field emission scanning electron microscope (FESEM), and the element content was determined through an energy dispersive X-ray (EDX) spectroscopy attached to the SU8010 FESEM (Hitachi, Tokyo, Japan). The morphology and structures of Au products were also captured and analyzed on a JEOL 2100 transmission electron microscope (TEM) (Tokyo, Japan). X-ray diffraction (XRD) pattern of porous Au worms was measured using a Bruker D8 Advance X-ray diffractometer (Cu irradiation, λ = 0.154056 nm) (Germany) within 2θ range of 30°–90°. ^1^H NMR and ^13^C NMR spectra of the synthesized [N_3333_][Gly] were performed on a Bruker Avance-400 NMR spectrometer (Billerica, MA USA) operating at 400.13 MHz at room temperature. A Necolet Nexus spectrometer (Thermo Fisher Scientific, USA) with KBr pellets was used to record Fourier transform infrared (FTIR) spectra in wavelength range from 400 cm^−1^ to 4000 cm^−1^. A Netzsch STA449C thermal analyzer (Erich NETZSCH GmbH & Co. Holding KG, Selb, Germany) was applied to perform the thermogravimetric (TG) measurements in air atmosphere at a heating rate of 10 °C min^−1^ in the temperature range of room temperature −600 °C.

### 2.4. SERS Measurements

For SERS sensitivity detection, R6G was chosen to serve as a probe molecule. First, porous worm-like Au samples dispersed in ethanol were dropped onto silicon wafer to create an even thin film by drying in atmosphere. Hence, the silicon wafers supporting sample film were buried in ethanol solution of R6G with different concentrations for 24 h. After that, washing them with ethanol and then drying were achieved in sequence. The SERS measurements were carried out using a HORIBA JobinYvon XploRA spectrophotometer (Kyoto, Japan) at room temperature with the laser wavelength of 532 nm. Similarly, *p*-aminothiophenol (PATP) and pyridine were also selected as probe molecules, respectively, for the SERS studies.

### 2.5. Catalytic Reduction of Nitaromatics

Typically, *p*-nitrophenol (*p*-NP) aqueous solution (1 mL, 0.42 mM), deionized water (0.24 mL), and NaBH_4_ aqueous solution (1.26 mL, 100 mM) were mixed in a quartz cuvette. Then, 0.5 mL of 5 mM porous worm-like Au aqueous solution was also put into the quartz cuvette. Then, the reduction reaction was measured using a TU-1900 UV–VIS spectrometer (Shimadzu, Kyoto, Japan) at room temperature. Similarly, the catalytic activity of porous worm-like Au for *p*-nitroaniline (*p*-NA) reduction was also evaluated with *p*-NA instead of *p*-NP under the same conditions.

## 3. Results and Discussion

Here, [N_3333_][Gly] was used to direct the growth and assembly of porous worm-like Au, and its influence on product shapes and structures was studied in detail. The structure of [N_3333_][Gly] was displayed in Figure 1. The results of ^1^H NMR (Appendix A), ^13^C NMR (Appendix A) and FTIR spectra (Appendix A) confirmed that the IL [N_3333_][Gly] with high purity could be obtained by using the current synthetic protocol and post-treatment procedure.

In the current protocol, porous worm-like Au could be easily obtained by one-pot route in aqueous solution under the modulation of [N_3333_][Gly] at room temperature. Its morphology and structure were detailedly characterized and analyzed. Figure 2 showed the field emission scanning electron microscope (FESEM) images of porous worm-like Au. From the low magnified overview Figure 2a,b, one could clearly see that the as-obtained Au samples were uniform in a large scale and represent worm-like structures with the length of 0.3–0.6 µm and the width of 100–150 nm. The high magnified FESEM images in Figure 2c,d showed that porous worm-like Au were composed of lots of small nanoparticles (NPs) with size of 6–12 nm. The NPs had very rough surfaces (see Figure 2b). These results revealed that the well-defined worm-like structures were orderly assembled by the generated Au NPs.

Figure 3a represented the corresponding transmission electron microscope (TEM) image of porous worm-like Au, from which one could obviously observe that porous worm-like Au consisted of many small NPs, which agreed well with the FESEM observation (Figure 2). Furthermore, large quantities of pores with various sizes were presented between NPs in Au worms, clearly illustrating the porous features of the as-obtained Au worms. Figure 3b showed high resolution transmission electron microscope (HRTEM) image of a NP as subunit in a porous Au worm. Its lattice spacing was determined to be 0.235 nm, attributed to the (111) facet of fcc structure of Au. The selected area electron diffraction (SAED) pattern forcefully confirmed the polycrystalline nature of porous Au worms (the inset of Figure 3b). In addition, twin boundary, amorphous state and edge atoms were also observed from Figure 3c,d. These crystal defects would act as active sites for highly efficient catalysis due to coordinative unsaturation. The X-ray diffraction (XRD) pattern of porous worm-like Au was shown in Appendix A, from which five characteristic peaks, located at 28.2°, 44.4°, 64.7°, 77.7° and 81.8°, could be obviously observed, ascribing to (111), (200), (220), (311) and (222) planes of the fcc structure of Au (JCPDS no. 04-0784), respectively. The impurity could not be found, confirming high purity and good crystallinity of porous Au worms. The chemical composition of porous Au worms was also analyzed by energy dispersive X-ray (EDX) spectroscopy. As displayed in Appendix A, only Au signals were presented, demonstrating the high purity of porous Au worms. In addition, in order to further examine if there is the existence of ILs and other impurity on the surface of the samples, FTIR and TG measurements of the porous Au worms were also achieved (Appendix A). It was found from Appendix A that FTIR spectra of the porous Au worms displayed a smooth curve in the studied wavenumber range and no perceptible characteristic absorption peak of ILs and other impurity could be found. On the other hand, TG measurement of the porous Au worms showed about 0.76% loss of weight at the temperature below 300 °C (Appendix A), which could be ascribed to the evaporation of water absorbed on the sample. Therefore, FTIR and TG results together with XRD and EDX analysis verified that the ILs and other impurities could be removed completely by using the current post-treatment procedure and the surface-clean porous Au worms could be obtained.

Figure 4a displayed N_2_ adsorption-desorption isotherms of porous Au worms. It can be seen that it took shape as a distinct type IV curve with hysteresis loop, indicating the presence of mesopores in the sample and that most pores existed in the interior of the sample than on the surface [33]. Its BET specific surface area is 71.16 m^2^ g^−1^, which is significantly larger than most of Au products with other shapes, such as mesoporous Au nanoleaves (2.38 m^2^ g^−1^) [34], mesoporous Au sponge (11.9 m^2^ g^−1^) [35], nanoporous Au foams (12 m^2^ g^−1^) [36], and 3D nanoporous Au (3.7 m^2^ g^−1^ [37] and 22.77 m^2^ g^−1^ [26]). The dramatically high surface area endowed porous Au worms with superior properties in most applications including catalysis and sensing. It was observed from Figure 4b that the pore sizes of porous Au worms had a broad distribution from micropores to mesopores and were dominated by the mesopores with the size of about 2.8 nm. Thus, the high specific surface area and various pore sizes provided porous Au worms with more accessible active sites.

Diverse terms were implemented to research the factors affecting the shapes, structures, and properties of samples, including ILs species and concentrations, reaction temperature, and reducing agent amount. First, [N_1111_][Gly] and [N_4444_][Gly] were employed to inspect the effects of alkyl chain length in cations of ILs on the Au products. It was found that in the presence of [N_1111_][Gly], the porous structures were also formed, but they had non-uniform shape and size (Figure 5a,b). When [N_4444_][Gly] was used instead of [N_3333_][Gly], the Au spherical particles with diameter of 100–200 nm were formed from aggregation of many small NPs (Figure 5c,d). It can be seen that alkyl chain length in the ILs affected the morphology of Au products to a certain extent. This could be understood from the fact that the difference of alkyl chain length in ILs changed the interactions of ILs with Au species, steric effects, and the microstructures of ILs in solution, which, in turn, changed the reaction rate and assembly of as-generated NPs.

Next, the influences of IL concentration changes were also examined. For this purpose, the concentration of [N_3333_][Gly] in solution was set to be 50, 150 and 200 mM, respectively. With 50 mM of [N_3333_][Gly] in aqueous solution, the sample exhibited spherical structures with diameter of about 40–120 nm and rough surface (Figure 6a and the insert). When it came to the typical synthesis (100 mM [N_3333_][Gly]), the well-defined Au worms with porous structures were obtained (Figure 6b). As the [N_3333_][Gly] concentration was further increased to 150 and 200 mM, the shape and size of the products gradually became ununiform and some dispered NPs also appeared. At the same time, the porous features were weaker in comparison with porous Au worms (see Figure 6c,d). These results indicated that suitable IL concentration was also vital to the formation of porous worm-like structures.

As well known, reaction temperature has significant effect on the reaction kinetics and then the morphology of products. When the reaction was proceeded at 7 °C, larger particles with irregular shapes were aggregated by the as-generated Au NPs (Figure 7a). In the typical reaction at room temperature (about 25 °C), the well-defined porous worm-like Au was assembled by small NPs (Figure 2). As the temperature rose to 40 °C, the size of Au sample decreased, but its uniformity dropped (Figure 7b). Further raising the temperature to 60 and 80 °C, the aggregates became smaller. At the same time, some dispersed NPs also appeared (Figure 7c,d). These results indicated that the reaction temperature surely had obvious influence on the shapes and sizes of products. With the increase of reaction temperature, the size of aggregates gradually decreased and the well-defined morphology became deteriorate, which were in good agreement with the observation [38]. At low temperature, the reaction was slow and less nuclei were formed. Under the synergistic action of [N_3333_]^+^ and [Gly]^−^ in ILs and their formed microstructures in solution, these nuclei began to grow and assembly orderly and formed the well-defined porous Au worms. However, at high temperature, the reaction was fast and a large number of NPs were generated at the initial stage. They moved quickly in solution and the interaction of them with ILs were weaken. At the same time, the microstructures of ILs in solution became disordered or even be disrupted at high temperature, resulting in the formation of smaller spherical aggregates and dispersed NP mixtures [39,40]. Thus, room temperature (about 25 °C) was suitable for the growth and assembly of the intriguing porous Au worms in the current case.

In addition, the amount of reductant ascorbic acid also affected the reduction rate and the morphology of the products. Compared to porous worm-like Au, the Au samples were aggregated by larger NPs with a lesser amount of ascorbic acid (Appendix A), while the smaller NPs were the subunits of aggregates with the greater amount of ascorbic acid (Appendix A). With the addition of less ascorbic acid, fewer Au nuclei (acting as seeds for the growth of other Au species) were formed at the beginning stage, leading to the formation of larger NPs. However, when more ascorbic acid was applied, more nuclei were generated and a large number of Au precursors could be expended, which led to the formation of the smaller NPs. At the same time, the amount of ascorbic acid also affected the assembly of the as-generated NPs. It could be seen from Appendix A that an unoptimizable amount of ascorbic acid would result in the aggregation of NPs into large and irregular structures. Only with suitable amount of reductant ascorbic acid and the subtle balance of reduction and growth kinetics, porous worm-like Au could be orderly assembled [41].

To illuminate the growth mechanism of porous Au worms, time-dependent experiments were carried out to trace their morphological evolution. As displayed in Figure 8a and the inset, Au spherical NPs with diameters of about 5 nm were formed at 3 min into the reaction. As the reaction extended to 10 min, the NPs became larger and rougher and took shapes as small stars with several petals grown on them (Figure 8b). When the reaction reached 4 h, the aggregates with loose structures were formed with small NPs as building blocks. Concomitantly, some dispersed NPs still existed (Figure 8c). Further prolonging the reaction time to 8 h, the porous and worm-like structures were observed (Figure 8d). Based on the aforementioned results, the formation of porous Au worms could be devide into three stages: (i) the nucleation and growth into small spherical NPs, (ii) further growth of spherical NPs into star-like NPs, and (iii) ordered assembly of stars into well-defined porous worms.

In view of the above experiments and analysis, the growth mechanism of well-defined porous Au worms could be deduced as follows. In aqueous [N_3333_][Gly] solution, AuCl_4_^−^ first matched with [N_3333_]^+^ via electrostatic interaction. After the fast addition of ascorbic acid, Au^3+^ was reduced to Au^0^, which gradually grew into nanostars under the modulation of [Gly]^−^ [42]. Hence, the NH_2_ and COO^−^ groups of [Gly]^−^ would interact with the Au nanostars [42,43,44,45], leading to the aggregation of nanostars via inter-NP bridges in [Gly]^−^ through binding two adjacent NPs [42]. At the same time, the [N_3333_]^+^ was also adsorbs on the surface of Au aggregation through electrostatic interaction with COO^−^ groups in [Gly]^−^. Thus, under the suitable concentration of [N_3333_][Gly] in aqueous solution, some ILs began to aggregate and take special shape, which, along with their interaction with Au NPs, directed the growth and ordered assembly of porous Au worms. When the reaction was performed under vigorous stirring, big bulks were obtained, which were formed by the aggregation of irregular particles (Appendix A). In the absence of [N_3333_][Gly], solid nanospheres with about 50 nm in diameter were the main products (Appendix A). In addition, as stated above, smaller spherical aggregates as well as dispersed NPs were formed at raised temperature. These control experiments showed that these products had obvious difference in morphology compared with porous Au worms, clearly illuminating that both the microstructures of [N_3333_][Gly] in aqueous solution and the interaction of [N_3333_][Gly] with Au were two key factors accounting for the growth and assembly of the well-defined porous Au worms.

SERS has been gaining great interest due to its remarkable signal sensitivity in target products, and has been deemed to be an effective mean for the test and analysis of traces involving chemistry, biochemistry and environment [6,46,47]. Due to well biocompatibility, high stability, and high SERS sensitivities, various Au nanostructures are frequently used as the SERS substrates [48]. Furthermore, those with rough surfaces, porous structures, and aggregation states, where abundant “hot spots” can be created at their edges, corners, gaps, and junctions, generally exhibit remarkable SERS enhancement [49,50].

To probe SERS sensitivity, R6G was chosen to serve as a model molecule. Figure 9a presented the SERS spectrum of R6G bound on porous Au worms at different concentrations. It can be seen that the intensity of Raman absorption bands of R6G on porous Au worms are obviously enhanced in comparison with the normal Raman spectrum of solid R6G (Appendix A). At the same time, several SERS peaks mildly moved to higher wavelengths, such as from 607, 769, 1357, and 1640 to 610, 771, 1358, and 1646 cm^−1^, correspondingly. The characteristic absorption bands are consistent well with those in literature [24,51,52]. Furthermore, as shown in Figure 9a, even if the concentration of R6G was as low as 10^−13^ M, the major bands could also be clearly identified, suggesting that with porous Au worms as SERS active substrate, the R6G detection limit was below 10^−13^ M. This detection limit was also obviously lower than those obtained on island-like nanoporous Au (10^−10^ M) [53], the nanoplate-built Au flowers (10^−11^ M) [32], and Ag nanosheet-assembled film (10^−12^ M) [54]. According to the equation S1 and S2, the enhancement factor (EF) of porous Au worms was calculated to be about 3.5 × 10^6^ (see Supporting Information) [32,52].
EF = (*I_SERS_*/*N_ads_*)/(*I_bulk_*/*N_bulk_*)(S1)
where *I_SERS_* and *I_bulk_* denote the Raman intensity of R6G in SERS and normal Raman spectrum, respectively. *N_ads_* and *N_bulk_* are the number of R6G molecules adsorbed on the SERS substrates and number of bulk molecules within the SERS detecting spot, respectively.

*N_ads_* could be calculated using the following equation:*N_ads_* = *N_d_A_laser_ A_N_*/σ(S2)
where *N_d_* is the number density of porous Au worms, *A_laser_* is the area of the focal spot of laser, *A_N_* is the footprint area of Au product, and σ is the surface area occupied by an adsorbed R6G molecule on full coverage of Au. For more details, please see Supporting Information.

Another important factor affecting the SERS applications was the reproducibility. To this end, five randomly chosen positions in porous worm-like Au film were measured and are shown in Figure 9b. It could be clearly observed that the SERS spectrum of R6G on the five positions was very similar, with less than 2% average relative standard deviations (calculated by the peak intensity at 610, 1358, and 1646 cm^−1^), demonstrating the extremely high reproducibility.

In addition, *p*-aminothiophenol (PATP) and pyridine were also used as test probe molecules to further examine the SERS sensitivities of porous Au worms. Figure 9c displayed the SERS spectra of 10^−6^ M PATP adsorbed on the porous Au worms, whose peaks were well consistent with those reported in literature [25,55,56]. It can be seen that the porous Au worms exhibited not only remarkable SERS responses but also high reproducibility with the average relative standard deviations of about 3% (calculated by the peak at 1576 cm^−1^). At the same time, as shown in Figure 9d, when pyridine was used as a probe molecule, the main characteristic peaks could still be found at the concentration of 10^−3^ M, confirming that the detection limit of pyridine on porous Au worms was lower than 10^−3^ M. Thus, the as-prepared porous worm-like Au presented remarkable SERS sensitivity and superior reproducibility. This, together with the inherent high stability of Au, makes the porous Au worms an ideal active substrate for the promising SERS applications.

Nitroaromatics are important chemical raw materials in industry for the production of medicines, dyes, perfumes, and others. However, due to the high solubility in water and accumulation in soil, they can cause serious pollution to the environment. Moreover, most nitroaromatics can irritate human eyes and threaten the health of people [57]. Thus, it is necessary to transform nitroaromatics into high value-added or less harmful matters. Here, *p*-nitrophenol (*p*-NP) and *p*-nitroaniline (*p*-NA) reduction were selected as model reactions to research the catalytic activities of porous Au worms. A UV–VIS spectrometer was applied to detect the reduction processes. Throughout the reduction reaction, as concentration of the reducing agent NaBH_4_ was much larger than that of nitroaromatics, the reactions were assumed to be applicable to the first-order kinetic reaction [58,59,60,61].

As well known, *p*-NP in water has an apparent peak at 317 nm, but it moves to 400 nm since *p*-nitrophenolate ions are formed after the addition of NaBH_4_ [62,63,64,65]. Figure 10a exhibited the absorption spectrum of *p*-NP reduction as a function of reaction time with porous Au worms as catalyst at ambient temperature. It was obviously noted that in the presence of porous Au worms, the intensity of 400 nm peak declined quickly as the reaction progressed. Simultaneously, due to the formation of *p*-aninophenol, a new peak at 315 nm appeared and its strength increased correspondingly. Notably, *p*-NP could be completely converted to *p*-aminophenol within only 100 s, which was 8.4 times faster than that achieved with 3D nanoporous Au as catalyst [26] and 1.5 times faster than cubic gold nanorattles [58]. Figure 10b showed the relationship of –ln(A/A_0_) of 400 nm versus reaction time with porous Au worms as catalyst, where A and A_0_ were the peak intensity of p-NP at the moment of t and 0, respectively. Experimental result indicated that the reduction could not occur without catalyst. However, using porous Au worms as catalyst, the reaction could rapidly proceed. Using the kinetic equation: –ln(A/A_0_) = kt, the reaction rate constant (k) for the reduction of *p*-NP was calculated to be 2.62 × 10^−2^ s^−1^. This value is obviously higher than that obtained with 3D nanoporous Au as catalyst (0.1903 min^−1^) [26]. The high activity of porous Au worms might be attributed to the various porous sizes, abundant crystal defects, large specific surface area, and a large number of active sites which led to widely exposed and accessible active sites for the noticeable boosts of catalytic efficiency [1].

Similarly, the catalytic reduction of *p*-NA was also achieved in the presence of porous Au worms. As well known, *p*-NA has an inherent absorption at 385 nm. In the absence of porous Au worms, this peak was almost not changed after the addition of NaBH_4_ without, indicating that the reduction of *p*-NA could not be progressed. Nevertheless, using porous worm-like Au as catalyst, the reaction could be quickly completed within about 150 s. Simultaneously, there appeared a new absorption band at 308 nm due to the generation of *p*-phenylenediamine (Figure 11a). Figure 11b exhibited the plot of –ln(A/A_0_) versus reaction time, from which k was calculated to be 1.38 × 10^−2^ s^−1^. This result indicated that for the reduction of *p*-NA, porous worm-like Au also demonstrated superior catalytic activity.

## 4. Conclusions

In summary, the well-defined porous worm-like Au can be facilely one-step synthesized under the modulation of IL [N_3333_][Gly], which exhibits the simple, green and low-cost features. It was shown that [N_3333_][Gly] is crucial to construct porous worm-like structures of Au. The as-prepared porous Au worms exhibited not only outstanding SERS sensitivity and superior reproducibility as SERS active substrate, but also excellent catalytic performance for the nitroaromatic reduction. We believe that this green and simple IL-assisted strategy can also be extended to grow and assemble other well-defined metallic nanostructures for valuable utilization.

## Figures and Tables

**Figure 1 nanomaterials-09-01772-f001:**
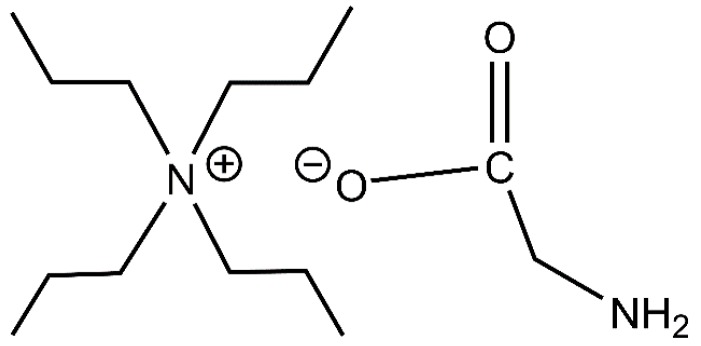
The structure of [N_3333_][Gly].

**Figure 2 nanomaterials-09-01772-f002:**
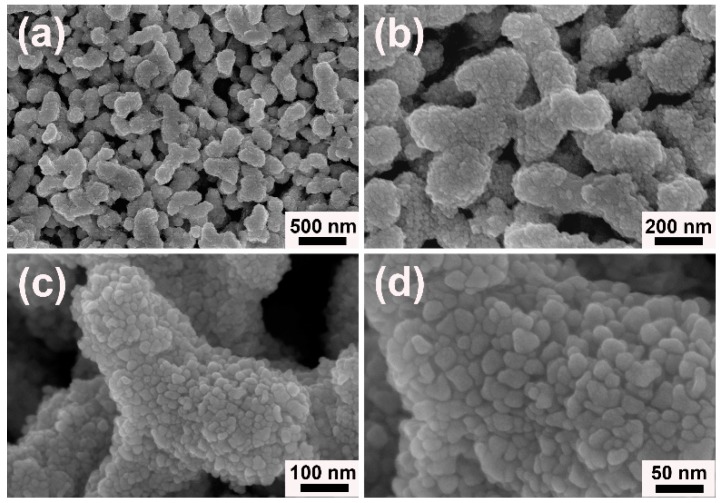
The differently magnified field emission scanning electron microscope (FESEM) images of porous worm-like Au synthesized in the presence of [N_3333_][Gly]: (**a**) 20,000 times, (**b**) 50,000 times, (**c**) 100,000 times and (**d**) 200,000 times.

**Figure 3 nanomaterials-09-01772-f003:**
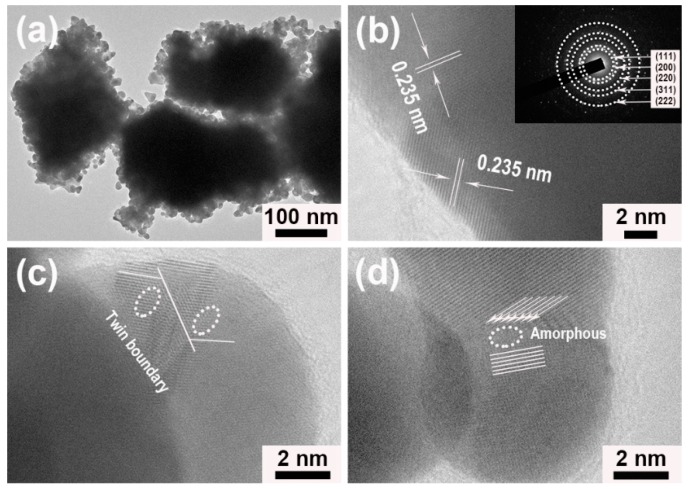
Porous Au worms synthesized in the presence of [N_3333_][Gly]: (**a**) transmission electron microscope (TEM) image, and (**b**–**d**) high resolution transmission electron microscope (HRTEM) images; the selected area electron diffraction (SAED) pattern is presented in the inset of (**b**).

**Figure 4 nanomaterials-09-01772-f004:**
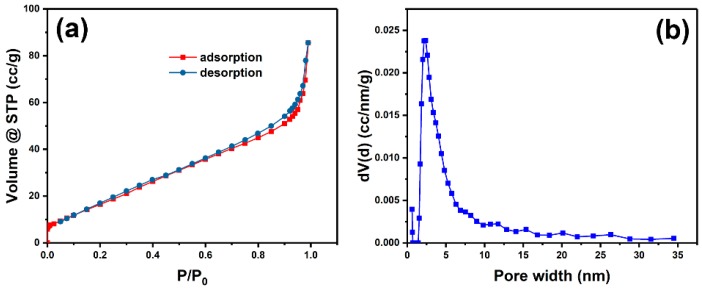
N_2_ adsorption-desorption isotherms (**a**) and porous size distribution (**b**) of porous Au worms.

**Figure 5 nanomaterials-09-01772-f005:**
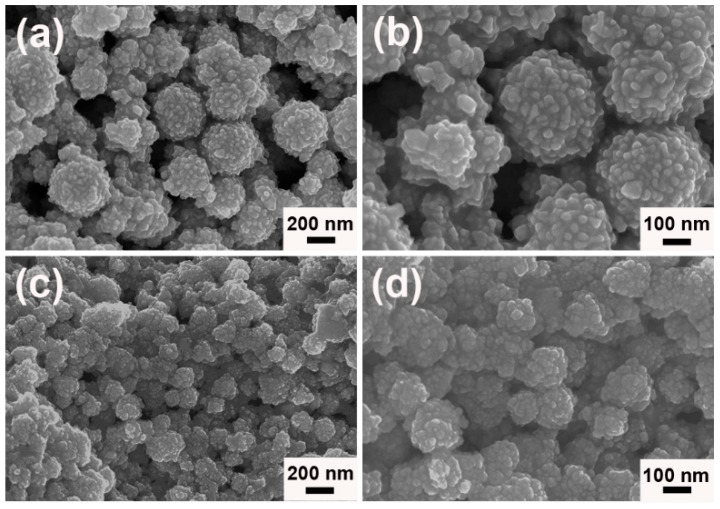
FESEM images of as-prepared Au samples assisted by different ILs: (**a**,**b**) [N_1111_][Gly], and (**c**,**d**) [N_4444_][Gly].

**Figure 6 nanomaterials-09-01772-f006:**
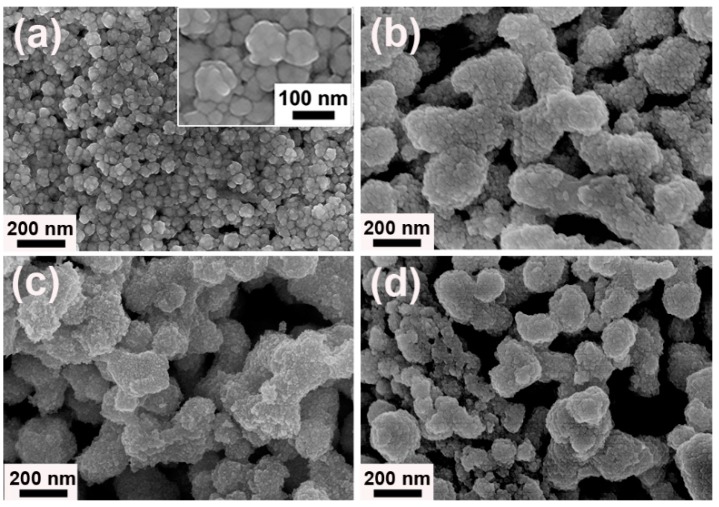
FESEM images of as-prepared Au samples assisted by [N_3333_][Gly] with various concentrations: (**a**) 50 mM, (**b**)100 mM, (**c**) 150 mM, and (**d**) 200 mM; the inset of (**a**) shows the magnified SEM image.

**Figure 7 nanomaterials-09-01772-f007:**
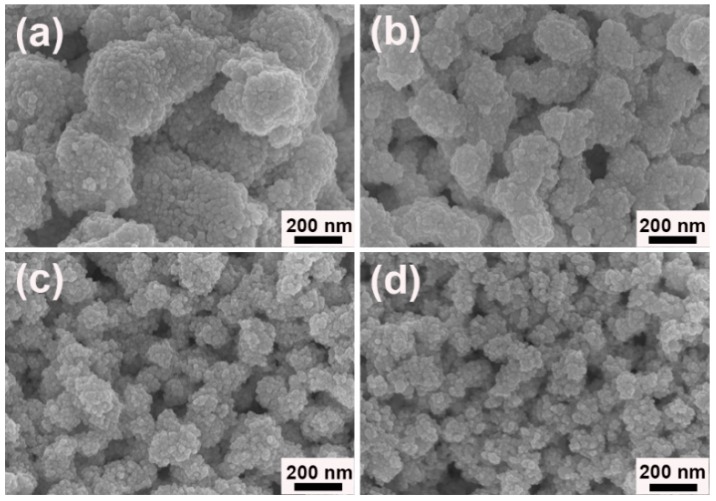
FESEM images of as-prepared Au samples assisted by [N_3333_][Gly] at different temperatures: (**a**) 7 °C, (**b**) 40 °C, (**c**) 60 °C, and (**d**) 80 °C.

**Figure 8 nanomaterials-09-01772-f008:**
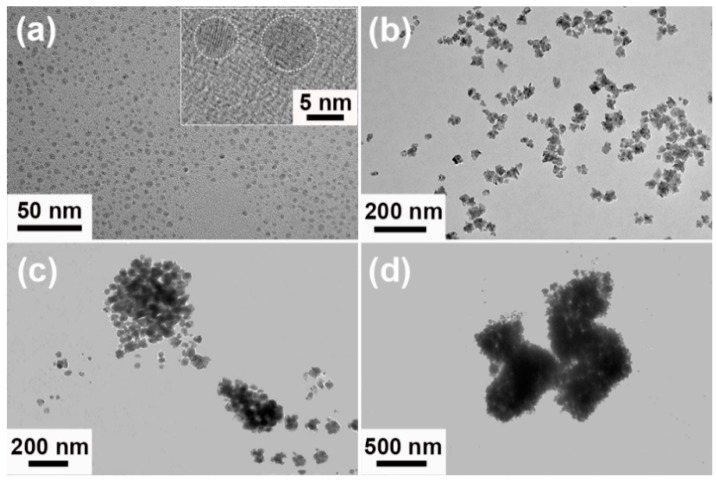
TEM images of as-synthesized Au products assisted by [N_3333_][Gly] at different interval: (**a**) 3 min, (**b**) 10 min, (**c**) 4 h, and (**d**) 8 h; the inset of (**a**) shows the magnified TEM image.

**Figure 9 nanomaterials-09-01772-f009:**
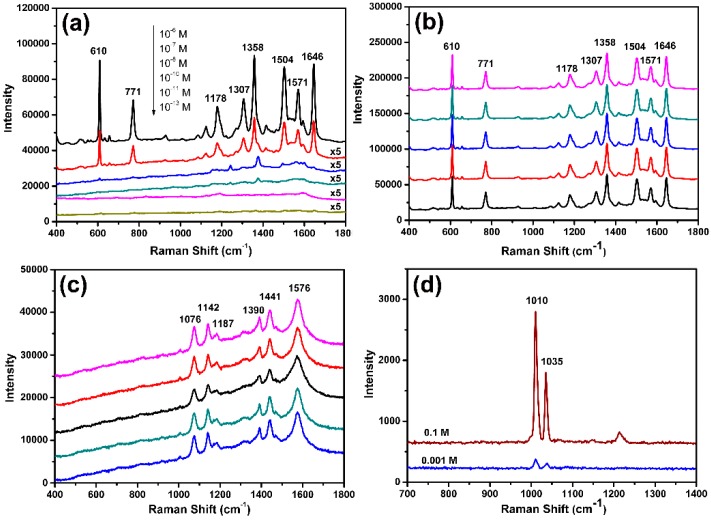
(**a**) SERS spectra of R6G bound on porous Au worms at different concentrations, (**b**) SERS spectra on five randomly chosen positions in porous worm-like Au film with 10^−6^ M R6G, (**c**) SERS spectra on five randomly chosen positions in porous worm-like Au film with 10^−6^ M PATP, and (**d**) SERS spectra of pyridine bound on porous Au worms at different concentrations.

**Figure 10 nanomaterials-09-01772-f010:**
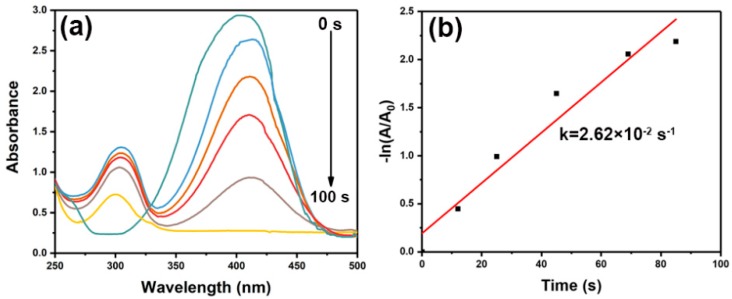
*P*-NP reduction reaction with porous Au worms as catalyst: (**a**) absorption spectrum as a function of time, and (**b**) the plot of –ln(A/A_0_) versus reaction time.

**Figure 11 nanomaterials-09-01772-f011:**
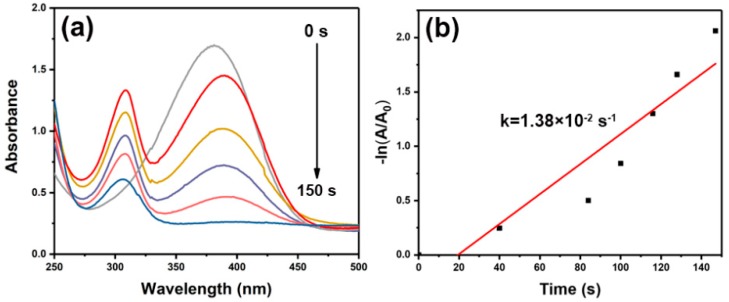
*P*-NA reduction reaction with porous Au worms as catalyst: (**a**) absorption spectrum as a function of time, and (**b**) the plot of −ln(A/A_0_) versus reaction time.

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
