# Peer review of "Ionic Liquid-Modulated Synthesis of Porous Worm-Like Gold with Strong SERS Response and Superior Catalytic Activities"

_nanomaterials, 2019, doi:10.3390/nano9121772_

Round 1

Reviewer 1 Report

This manuscript reports on a green and simple process for the growth of porous worm-like Au in aqueous solution at room temperature. The resulting worm-like Au structures demonstrated outstanding SERS response and were evaluated in terms of catalytic performance for the reduction of nitroaromatics in water. The manuscript has a sound and explicit experimental section with self-explanatory SEM and TEM characterization images. Also, the growth mechanism is discussed. However, it is not clear if the reported Au growth process is employed for the first time in the literature, so the authors should shed light on this point. Other minor issues which should be addressed before the paper is accepted for publication in nanomaterials are the following:

Please revise English extensively. Syntax and grammar errors are present across the text, e.g. Introduction lines 43-44 please use past tense consistently; Introduction line 67, “represented to grow” should be changed to “employed for the growth” Please discuss further the effect of the synthesis temperature on the growth morphology and how can temperature affect the proposed mechanism.

Author Response

Referee 1:

This manuscript reports on a green and simple process for the growth of porous worm-like Au in aqueous solution at room temperature. The resulting worm-like Au structures demonstrated outstanding SERS response and were evaluated in terms of catalytic performance for the reduction of nitroaromatics in water. The manuscript has a sound and explicit experimental section with self-explanatory SEM and TEM characterization images.

Author’s response: Thanks for the comments.

Comment 1:

Also, the growth mechanism is discussed. However, it is not clear if the reported Au growth process is employed for the first time in the literature, so the authors should shed light on this point.

Author’s response: Thanks for the comments. According to the Reviewer’s suggestion, we fully research the relevant literature and find that there are only a few reports involving tetraalkylammonium amino acid-based ILs. Most of them focus on the absorption of CO2 [1-6], surface thermodynamics [7] and the phase-forming ability in aqueous two-phase systems (ATPSs) [8]. Only Li et al use [N1111]Asn as template and prepare the core-shell structural surface imprinting microspheres (MIPs) via surface-initiated reversible addition-fragmentation chain transfer (RAFT) polymerization in acetonitrile/ethanol solvent at 70 ℃ [9]. Moreover, none of them uses the IL [N3333][Gly] in their studies. In addition, Au NPs have been prepared with the assistance of amino acids [10-13], but they have obvious differences in morphology with porous Au worms in our work. Thus, to the best of our knowledge, the growth process of porous Au worms is proposed for the first time.

Based on the analysis mentioned above, the sentence “To the best of our knowledge, it is the first time that tetraalkylammonium amino acid-based ILs was utilized to assist the growth of the special Au nanostructures and their growth process was proposed.” has been added in the revised manuscript (page 2, lines 75-77).

Comment 2:

Other minor issues which should be addressed before the paper is accepted for publication in nanomaterials are the following:

Please revise English extensively. Syntax and grammar errors are present across the text, e.g. Introduction lines 43-44 please use past tense consistently; Introduction line 67, “represented to grow” should be changed to “employed for the growth”

Author’s response: Thanks for the comments. According to the Reviewer’s suggestion, the sentences in Introduction lines 43-44 and others in the original manuscript have been revised into past tense consistently in the revised manuscript. The phrase “represented to grow and assemble” (page 2, line 67 in original manuscript) has been replaced by “employed for the growth and assembly of” in the revised manuscript (page 2, line 72).

In addition, we have invited a professional guidance to careful review our manuscript and give some valuable revisions. Then, we carefully examine our manuscript once again.

Therefore, the following revisions have also been made in the revised manuscript:

(1) The inaccurate words and phrases “catalytic activities” (title in page 1, line 3 in the original manuscript), “A typical method for synthesizing ionic liquids” (page 2, line 83 in the original manuscript), “The process for synthesis of worm-like Au. Typically,” (page 2, line 92 in the original manuscript), “gold” (page 3, line 127; page 4, line 145; page 4, line 151; page 8, line 273 in the original manuscript), “The high magnified images” (page 3, line 132 in the original manuscript), “was consisted of ” (page 4, line 141 in the original manuscript), “porous worm-like Au” (page 4, line 144 in the original manuscript), “porous worm-like Au” (page 4, line 153; page 4, line 158; page 6, line 205; page 7, line 222 in the original manuscript), “worm-likes” (page 4, line 155 in the original manuscript), “under the modulation of” (page 4, line 158 in the original manuscript), “However” (page 6, line 204 in the original manuscript), “assembly” (page 7, line 235 in the original manuscript), “took the shape” (page 7, line 243 in the original manuscript), “of” (page 8, line 259 and 260 in the original manuscript), “test” (page 8, line 271 in the original manuscript), “occurred” (page 9, line 321 in the original manuscript), and “versus reaction Time” (page 10, line 332 in the original manuscript) have been replace by “Catalytic Activities” (title in page 1, line 3 in the revised manuscript), “For a typical synthesis of ionic liquid” (page 2, line 91 in the revised manuscript), “For the typical synthesis of porous worm-like Au” (page 2, line 100 in the revised manuscript), “Au” (page 4, line 155; page 4, line 173; page 4, line 179; page 8, line 418 in the revised manuscript), “The high magnified FESEM images” (page 4, line 160 in the revised manuscript), “consisted of ” (page 4, line 169 in the revised manuscript), “a porous Au worm” (page 4, line 172 in the revised manuscript), “porous Au worms” (page 4, line 181; page 5, line 241; page 6, line 302; page 7, line 342 in the revised manuscript), “worms” (page 4, line 183 in the revised manuscript), “in the presence of” (page 5, line 241 in the revised manuscript), “At the same time” (page 6, line 301 in the revised manuscript), “the assembly” (page 7, line 354 in the revised manuscript), “took shapes” (page 7, line 362 in the revised manuscript), “in” (page 8, lines 402 and 404 in the revised manuscript), “the test” (page 8, line 416 in the revised manuscript), “occur” (page 10, line 551 in the revised manuscript), and “versus reaction time” (page 10, line 561 in the revised manuscript), respectively.

(2) The words “Fig.” in the original manuscript have been replaced by “Figure” throughout the revised manuscript.

(3) The words “(Figure 5c and d)” have been added in the revised manuscript (page 5, line 264), and the words “further” (page 4, line 154 in the original manuscript) and “and accessible” (page 9, line 326 in the original manuscript) have been deleted in the revised manuscript.

(4) The sentences (page 7, lines 212-217 in the original manuscript) “It is known that reaction temperature also has significant effect on the reaction kinetics and then morphology of the products. When the reaction was proceeded at 7 ℃, larger particles with irregular shapes were aggregated by the as-generated Au NPs (Figure 7a). In the typical reaction at room temperature (about 25 ℃), the well-defined porous worm-like Au was assembled by small NPs (Figure 1). As the temperature was rised to 40 ℃, the size of Au samples decreased, but their uniformity was droped down (Figure 7b).” have been changed into “As well known, reaction temperature has significant effect on the reaction kinetics and then the morphology of products. When the reaction was proceeded at 7 ℃, larger particles with irregular shapes were aggregated by the as-generated Au NPs (Figure 7a). In the typical reaction at room temperature (about 25 ℃), the well-defined porous worm-like Au was assembled by small NPs (Figure 1). As the temperature was rised to 40 ℃, the size of Au sample decreased, but its uniformity was droped down (Figure 7b).” in the revised manuscript (page 7, lines 324-329).

(5) The sentence (page 9, lines 304-305 in the original manuscript) “Moreover, most nitroaromatics can irritate human eyes, then threatening health of peoples [52].” has been changed into “Moreover, most nitroaromatics can irritate human eyes and then threaten the health of peoples [57].” in the revised manuscript (page 10, lines 533-534).

(6) The sentence (page 10, lines 334-336 in the original manuscript) “As well known, p-NA has an inherent absorption at 385 nm which was almost not changed after the addition of NaBH4 without porous Au worms, indicating that the reduction of p-NA could not be progressed.” has been changed into “As well known, p-NA has an inherent absorption at 385 nm. In the absence of porous Au worms, this peak was almost not changed after the addition of NaBH4, indicating that the reduction of p-NA could not be progressed.” in the revised manuscript (page 10, lines 563-564).

(7) The sentence (page 10, lines 347-348 in the original manuscript) “It was shown that [N3333][Gly] is crucial to the construction of porous Au worms.” has been changed into “It was shown that [N3333][Gly] is crucial to construct the porous worm-like structures of Au.” in the revised manuscript (page 11, lines 594-595).

(8) The font and font size of the captions in Figure 11 have changed into Palatino linotype and 10, respectively.

Comment 3:Please discuss further the effect of the synthesis temperature on the growth morphology and how can temperature affect the proposed mechanism. 

Author’s response: Thanks for the comment. According to the Reviewer’s suggestion, the effect of the synthesis temperature on the growth morphology was further discussed. As shown in the original manuscript (page 7, paragraph 1 and Figure 7), when the reaction was proceeded at 7 ℃, larger particles with irregular shapes were aggregated by the as-generated Au NPs. In the typical reaction at room temperature (about 25 ℃), the well-defined porous worm-like Au was assembled by small NPs. As the temperature was rised to 40 ℃, the size of Au samples decreased, but its uniformity was droped down (Figure 7b). Further raising the temperature to 60 ℃ and 80 ℃, the aggregates became smaller. At the same time, some dispersed NPs also appeared. These results indicated that the reaction temperature had obvious influence on the shapes and sizes of products. With the increase of reaction temperature, the size of aggregates gradually decreased and the well-defined morphology became deteriorate, which were in good agreement with the observation [14]. At low temperature, the reaction was slow and less nuclei were formed. Under the synergistic action of [N3333]+ and [Gly]- in ILs and their formed microstructures in solution, these nuclei began to grow and assembly orderly and formed the well-defined porous Au worms. However, at high temperature, the reaction was fast and lots of NPs were generated at the initial stage. They moved quickly in solution and the interaction of them with ILs were weaken. At the same time, the microstructures of ILs in solution became disordered or even be disrupted at high temperature, resulting in the formation of smaller spherical aggregates and dispersed NP mixtures [15,16]. Thus, room temperature (about 25 ℃) was suitable for the growth and assembly of the intriguing porous worm-like Au in current case.

Based on the aforementioned analysis, the following revisions have been made in the revised manuscript.

(1) The sentence (Page 7, lines 219-221 in the original manuscript) “which could be explained from the assumption that temperature change altered the reaction rate and solution microstructures of the ILs and then the particle growth and assembly.” have been deleted in the revised manuscript.

(2) The following sentences have been added into the revised manuscript (page 7, lines 332-340).

“With the increase of reaction temperature, the size of aggregates gradually decreased and the well-defined morphology became deteriorate, which were in good agreement with the observation [38]. At low temperature, the reaction was slow and less nuclei were formed. Under the synergistic action of [N3333]+ and [Gly]- in ILs and their formed microstructures in solution, these nuclei began to grow and assembly orderly and formed the well-defined porous Au worms. However, at high temperature, the reaction was fast and lots of NPs were generated at the initial stage. They moved quickly in solution and the interaction of them with ILs were weaken. At the same time, the microstructures of ILs in solution became disordered or even be disrupted at high temperature, resulting in the formation of smaller spherical aggregates and dispersed NP mixtures [39,40].”

(3) The sentence “In addition, as stated above, smaller spherical aggregates as well as dispersed NPs were formed at raised temperature” has been added into the revised manuscript (page 8, lines 409-410).

(4) The corresponding References have been added into the revised manuscript.

References

[1] Zhang, F.; Fang, C.-G.; Wu, Y.-T.; Wang, Y.-T.; Li, A.-M.; Zhang, Z.-B. Absorption of CO2 in the aqueous solutions of functionalized ionic liquids and MDEA. Chem. Eng. J. 2010, 160, 691-697.

[2] Zhang, F.; Ma, J.-W.; Zhou, Z.; Wu, Y.-T.; Zhang, Z.-B. Study on the absorption of carbon dioxide in high concentrated MDEA and ILs solutions. Chem. Eng. J. 2012, 181-182, 222-228.

[3] Jiang, Y.-Y.; Wang, G.-N.; Zhou, Z.; Wu, Y.-T.; Geng, J.; Zhang, Z.-B. Tetraalkylammonium amino acids as functionalized ionic liquids of low viscosity. Chem. Commun. 2008, 505-507.

[4] Fu, Dong.; Zhang, P.; Mi, C.L. Effects of concentration and viscosity on the absorption of CO2 in [N1111][Gly] promoted MDEA (methyldiethanolamine) aqueous solution. Energy 2016, 101, 288-295.

[5] Zhou, Z.; Jing, G.; Zhou, L. Characterization and absorption of carbon dioxide into aqueous solution of amino acid ionic liquid [N1111][Gly] and 2-amino-2-methyl-1-propanol. Chem. Eng. J. 2012, 204-206, 235-243.

[6] McDonald. J.L.; Sykora, R.E.; Hixon, P.; Mirjafari, A.; Jr, J.H.D. Impact of water on CO2 capture by amino acid ionic liquids. Environ. Chem. Lett. 2014, 12, 201-208.

[7] Xie, J.; Wang, F.; Fu, D. Investigation of surface thermodynamics for DEAE-[N1111][Gly], DEAE-[Bmim][Gly] and DEAE-[Bmim][Lys] aqueous solutions. J. Mol. Liq. 2018, 249, 1068-1074.

[8] Wu, C.; Wang, J.; Li, Z.; Jing, J.; Wang, H. Relative hydrophobicity between the phases and partition of cytochrome-c in glycine ionic liquids aqueous two-phase systems. J. Chromatogr. A 2013, 1305, 1-6.

[9] Li, J.; Hu, X.; Guan, P.; Song, R.; Zhang, X.; Tang, Y.; Wang, C.; Qian, L. Preparation of core-shell structural surface molecular imprinting microspheres and recognition of L-Asparagine based on [N1111]Asn ionic liquid as template. React. Funct. Polym. 2015, 88, 8-15.

[10] Plascencia-Villa, G.; Torrente, D.; Marucho, M.; José-Yacamán, M.; Biodirected synthesis and nanostructural characterization of anisotropic gold nanoparticles. Langmuir 2015, 31, 3527-3536.

[11] Zakaria, H.M.; Shah, A.; Konieczny, M.; Hoffmann, J.A.; Nijdam, A.J.; Reeves, M.E. Small molecule- and amino acid-induced aggregation of gold nanoparticles. Langmuir 2013, 29, 7661-7673.

[12] Wangoo, N.; Bhasin, K.K.; Mehta, S.K.; Suri, C.R. Synthesis and capping of water-dispersed gold nanoparticles by an amino acid: Bioconjugation and binding studies. J. Colloid Interface Sci. 2008, 323, 247-254.

[13] Pakiari, A.H.; Jamshidi, Z. Interaction of amino acids with gold and silver clusters. J. Phys. Chem. A 2007, 111, 4391-4396.

[14] Huang, X.; Li, Y.; Chen, Y.; Zhou, E.; Xu, Y.; Zhou, H.; Duan, X.; Huang, Y. Palladium-based nanostructures with highly porous features and perpendicular pore channels as enhanced organic catalysts. Angew. Chem. 2013, 125, 2580-2584.

[15] Lv, H.; Xu, D.; Henzie, J.; Feng, J.; Lopes, A.; Yamauchi, Y.; Liu, B. Mesoporous gold nanospheres via thiolate-Au(I) intermediates. Chem. Sci. 2019, 10, 6423-6430.

[16] Lv, H.; Sun, L.; Lopes, A.; Xu, D.; Liu, B. Insights into compositional and structural effects of bimetallic hollow mesoporous nanospheres toward ethanol oxidation electrocatalysis. J. Phys. Chem. Lett. 2019, 10, 5490-5498.

Reviewer 2 Report

The impurity content of the synthesized ionic liquids can significantly affect the morphology and structure of the materials obtained. Therefore, the authors have to add NMR and IR spectra of the synthesized ionic liquids to the manuscript.

Page 5, line 178, authors wrote: First, IL [N3333][OH] was selected instead of [N3333][Gly] to study the effect of anion nature. This idea was not good. [N3333] [OH] is not an IL but an aqueous solution. There is a high possibility that this solution absorbed a certain amount of CO2, the authors did not mention how they prevented it. If I understood the experimental procedure (page 3, line 93-95) correctly in the solution, the concentration of OH ions was two times higher than the concentration of HAUCl4. In this case, gold ions can't be translated into Al (OH) 4, quantitatively (page 5, line 182). The authors should exclude this section from the manuscript or synthesize IL with some other anion and demonstrate its impact.

After the addition of ascorbic acid, it is possible to protonate some of the glycinate anions to glycine. The authors should perform TG / DTA measurements for the material obtained and show that there are no residues of organic compounds.

Author Response

Referee 2:

Comment 1:

The impurity content of the synthesized ionic liquids can significantly affect the morphology and structure of the materials obtained. Therefore, the authors have to add NMR and IR spectra of the synthesized ionic liquids to the manuscript.

Author’s response: Thanks for the comments. According to the Reviewer’s suggestion, NMR and FTIR spectra of the synthesized [N3333][Gly] have been achieved and shown as follows.

Figure A 1H NMR spectra of synthesized IL [N3333][Gly].

1H NMR (400 MHz, D2O) δ 3.09 - 2.93 (m, 10H, H(1) and (4)), 1.62 - 1.46 (m, 8H, H(2)), 0.78 (t, J = 7.3 Hz, 12H, H(3)).

Figure B 13C NMR spectra of synthesized IL [N3333][Gly].

13C NMR (101 MHz, D2O) δ 181.15 (s, C(5)), 59.84 (s, C(1)), 44.53 (s, C(4)), 14.77 (s, (2)), 9.78 (s, C(3)).

Figure C FTIR spectra of (a) [N3333][Gly] and (b) porous Au worms.

Figure C(a) presents the FTIR spectra of synthesized [N3333][Gly]. The wide peak at 3381 cm-1 is attributed to -NH2 asymmetric telescopic vibration. The peaks at 2972 and 2879 cm-1 belong to asymmetric and symmetric stretching vibration of -CH3, respectively. The peak at 1570 cm-1 is the characteristic absorption of -COO-. The band at 1386 cm-1 is assigned to the symmetric variable angle vibration of -CH3. The 972 cm-1 is ascribed to the C-N bending vibration in quaternary ammonium ions.

The results confirm that the [N3333][Gly] with high purity can be obtained by using the current synthetic protocol and post-treatment procedure, and the impurity content in it is negligible.

Based on the results and analysis aforementioned, the following revisions have been made in the revised manuscript:

(1) The sentence “The results of 1H NMR (Figure S1), 13C NMR (Figure S2) and FTIR spectra (Figure S3) confirmed that the IL [N3333][Gly] with high purity could be obtained by using current synthetic protocol and post-treatment procedure.” has been added into the revised manuscript (page 3, lines 147-149).

(2) The sentences “1H NMR and 13C NMR spectra of the synthesized [N3333][Gly] were performed on a Bruker Avance-400 NMR spectrometer operating at 400.13 MHz at room temperature. A Necolet Nexus spectrometer with KBr pellets was used to record Fourier transform infrared (FTIR) spectra in wavelength range from 400 cm-1 to 4000 cm-1.” have been added into the revised manuscript (page 3, lines 121-125).

(3) The corresponding figures and their analysis have been added into Supporting Information.

Comment 2:

Page 5, line 178, authors wrote: First, IL [N3333][OH] was selected instead of [N3333][Gly] to study the effect of anion nature. This idea was not good. [N3333] [OH] is not an IL but an aqueous solution. There is a high possibility that this solution absorbed a certain amount of CO2, the authors did not mention how they prevented it. If I understood the experimental procedure (page 3, line 93-95) correctly in the solution, the concentration of OH ions was two times higher than the concentration of HAuCl4. In this case, gold ions can't be translated into Au(OH)4-, quantitatively (page 5, line 182). The authors should exclude this section from the manuscript or synthesize IL with some other anion and demonstrate its impact.

Author’s response: Thanks for the valuable comments. According to the Reviewer’s suggestion, the section of the [N3333][OH] effect on products has been deleted in the revised manuscript. The corresponding revisions have been achieved and shown as follows.

  (1) Figure 5 in the original manuscript has been replaced by Figure D.

Figure D FESEM images of as-prepared Au samples assisted by different ILs: (a, b) [N1111][Gly], and (c, d) [N4444][Gly].

(2) The sentences (page 5, lines 177-184 in the original manuscript) “IL [N3333][OH] was selected instead of [N3333][Gly] to study the effect of anion nature. It can be seen from Figure 5a and b that using [N3333][OH], the NPs about 20 - 40 nm in diameter were formed, and several of them were easy to aggregate. This indicates that the anion nature in ILs had important influence on the shapes and sizes. In the presence of [N3333][OH], the solution had a high pH value and Au3+ existed in the form of Au(OH)4-, which led to fast nucleation and growth rate [38]. In addition, [OH]- and [Gly]- in the ILs had different interactions with Au species, which also resulted in the changes of reaction rate, and in turn the differences in morphology and structures. Besides anions,” have been deleted in the revised manuscript.

(3) The word (page 5, line 184 in the original manuscript) “also” has been deleted in the revised manuscript.

(4) The sentence (page 5, lines 188-189 in the original manuscript) “Alkyl chain length in the ILs also affected morphology of Au products to a certain extent.” has been changed into “It can be seen that alkyl chain length in the ILs affected the morphology of Au products to a certain extent.” in the revised manuscript (page 5, lines 264 and 288 in the revised manuscript).

(5) The sentence (page 5, lines 192-193 in the original manuscript) “Thus, both anions and cations in the ILs had obvious influence on the formation of porous worm-like Au.” has been deleted in the revised manuscript.

Comment 3:

After the addition of ascorbic acid, it is possible to protonate some of the glycinate anions to glycine. The authors should perform TG/DTA measurements for the material obtained and show that there are no residues of organic compounds

Author’s response: Thanks for the comments. According to the Reviewer’s suggestion, TG measurements for the porous Au worms has been achieved. It can be seen from Figure E that about 0.77% loss of weight is observed from TG curve of the porous Au worms. The loss of weight mainly occurs below 300 ℃, which may be attributed to the evaporation of water absorbed on the samples. In addition, as displayed in Figure C(b), FTIR spectra of purified porous Au worms exhibits a smooth curve and no perceptible characteristic absorption peak of [N3333][Gly] and other impurity can be found in the studied wavenumber range. Therefore, FTIR and TG results together with XRD and EDX analysis (see page 4, lines 176-182 in the revised manuscript and Figure S4 in revised supporting information) verify that the ILs and other impurity can be removed completely by using the current post-treatment procedure and the surface-clean porous Au worms can be obtained.

Figure E TGA curve of porous Au worms.

According to the above results, the following revisions have been made in the revised manuscript.

(1) The following sentences have been added in the revised manuscript (page 4, lines 183-192):

“In addition, in order to further examine if there is the existence of ILs and other impurity on the surface of the samples, FTIR and TG measurements of the porous Au worms were also achieved (Figure S3 and S5). It was found from Figure S3b that FTIR spectra of the porous Au worms displayed a smooth curve in the studied wavenumber range and no perceptible characteristic absorption peak of ILs and other impurity could be found. On the other hand, TG measurement of the porous Au worms showed about 0.76% loss of weight at the temperature below 300 ℃ (Figure S5), which could be ascribed to the evaporation of water absorbed on the sample. Therefore, FTIR and TG results together with XRD and EDX analysis verified that the ILs and other impurity could be removed completely by using the current post-treatment procedure and the surface-clean porous Au worms could be obtained.”

(2) The sentence “A Netzsch STA449C thermal analyzer was applied to perform the thermogravimetric (TG) measurements in air atmosphere at a heating rate of 10 ℃ min-1 in the temperature range of room temperature - 600 ℃.” has been added into the revised manuscript (page 3, lines 125-127).

(3) The corresponding figures have been added into Supporting Information.

Reviewer 3 Report

In this work Yao et al. described synthesis of porous worm-like gold nanoclusters and presented two example applications of the obtained nanostructures: (i) for surface enhanced Raman scattering (SERS) measurements of rhodamine 6G and (ii) for catalytic reduction of para-nitrophenol. In my opinion all described experiments have been carried out correctly and the interpretation of the obtained results is also correct. Moreover, Yao et al. carefully characterized the morphology of the obtained nanostructures. Therefore, I recommend publication of this work. If it is possible, I would only like to suggest Authors, to test obtained material for SERS measurements of an analyte other than rhodamine 6G. Rhodamine 6G is an extremely efficient Raman scatterer, and, usually, if present in the analysed sample, the signal from rhodamine 6G dominates the measured SERS spectrum. Yao et al. synthesised gold nanocluters using relatively complex reagents and one can assume that the surface of the obtained nanoparticles is contaminated. Therefore, it would be interesting to verify whether the contribution from the possible impurities would be seen in SERS spectrum of the weak scatterer, like, for example, pyridine, which is the standard scatterer for SERS measurements.

Author Response

Referee 3:

In this work Yao et al. described synthesis of porous worm-like gold nanoclusters and presented two example applications of the obtained nanostructures: (i) for surface enhanced Raman scattering (SERS) measurements of rhodamine 6G and (ii) for catalytic reduction of para-nitrophenol. In my opinion all described experiments have been carried out correctly and the interpretation of the obtained results is also correct. Moreover, Yao et al. carefully characterized the morphology of the obtained nanostructures. Therefore, I recommend publication of this work.

Author’s response: Thanks for the comments.

Comment 1:

If it is possible, I would only like to suggest Authors, to test obtained material for SERS measurements of an analyte other than rhodamine 6G. Rhodamine 6G is an extremely efficient Raman scatterer, and, usually, if present in the analysed sample, the signal from rhodamine 6G dominates the measured SERS spectrum. Yao et al. synthesized gold nanocluters using relatively complex reagents and one can assume that the surface of the obtained nanoparticles is contaminated. Therefore, it would be interesting to verify whether the contribution from the possible impurities would be seen in SERS spectrum of the weak scatterer, like, for example, pyridine, which is the standard scatterer for SERS measurements.

Author’s response: Thanks for the comments. According to the Reviewer’s suggestion, p-aminothiophenol (PATP) and pyridine are also used as test probe molecules to further examine the SERS sensitivities of porous Au worms. Figure F(c) displays the SERS spectra of 10-6 M PATP adsorbed on the porous Au worms, whose peaks are well consistent with those reported in literature [1-3]. It can be seen that the porous Au worms exhibit not only remarkable SERS responses but also high reproducibility with the average relative standard deviations of about 3% (calculated by the peak at 1576 cm-1). In addition, as shown in Figure F(d), when pyridine is used as a probe molecule, the main characteristic peaks can still be found at the concentration of 10-3 M, confirming that the detection limit of pyridine on porous Au worms is lower than 10-3 M. At the same time, no others peak of impurity can be found throughout the studied wavenumber range, further demonstrating the high purity of porous Au worms.

Figure F (a) SERS spectra of R6G bound on porous Au worms at different concentrations, (b) SERS spectra on five randomly chosen positions in porous worm-like Au film with 10-6 M R6G, (c) SERS spectra on five randomly chosen positions in porous worm-like Au film with 10-6 M PATP, and (d) SERS spectra of pyridine bound on porous Au worms at different concentrations.

Based on the above analysis, the following revisions have been made in the revised manuscript.

(1) The following sentences have been added in the revised manuscript (page 9, lines 498-506).

“In addition, p-aminothiophenol (PATP) and pyridine were also used as test probe molecules to further examine the SERS sensitivities of porous Au worms. Figure 9c displayed the SERS spectra of 10-6 M PATP adsorbed on the porous Au worms, whose peaks were well consistent with those reported in literature [25, 55, 56]. It can be seen that the porous Au worms exhibited not only remarkable SERS responses but also high reproducibility with the average relative standard deviations of about 3% (calculated by the peak at 1576 cm-1). At the same time, as shown in Figure 9d, when pyridine was used as a probe molecule, the main characteristic peaks could still be found at the concentration of 10-3 M, confirming that the detection limit of pyridine on porous Au worms was lower than 10-3 M.”

(2) Figure 9 in the original manuscript has been replaced by Figure F and corresponding captions have been changed into “Figure 9 (a) SERS spectra of R6G bound on porous Au worms at different concentrations, (b) SERS spectra on five randomly chosen positions in porous worm-like Au film with 10-6 M R6G, (c) SERS spectra on five randomly chosen positions in porous worm-like Au film with 10-6 M PATP, and (d) SERS spectra of pyridine bound on porous Au worms at different concentrations.” in the revised manuscript (page 9).

(3) The sentence (page 2, lines 79-80 in the original manuscript) “Aladdin Chem. Co provided p-Nitrophenol (p-NP, 98%) and Rhodamine 6G (R6G, 95%).” has been changed into “Aladdin Chem. Co provided p-Nitrophenol (p-NP, 98%), Rhodamine 6G (R6G, 95%), p-aminothiophenol (PATP, 97%) and pyridine (99.5%).” in the revised manuscript (page 2, lines 87-89).

(4) The sentence “Similarly, p-aminothiophenol (PATP) and pyridine were also selected as probe molecules, respectively, for the SERS studies.” has been added into the revised manuscript (page 3, lines 135-136).

(5) The corresponding References have been added into the revised manuscript.

References

[1] Yao, K.; Li, Z.; Li, X.; Lu, W.; Xu, A.; Zhang, H.; Wang, J. Tunable synthesis of Ag films at the interface of ionic liquids and water by changing cationic structures of ionic liquids. Cryst. Growth Des. 2017, 17, 990-999.

[2] Yao, K.-S.; Zhao, H.-L.; Wang, N.; Li, T.-J.; Zhao, S.; Lu, W.-W. Facile synthesis of ultra-large, single-crystal Ag nanosheet-assembled films at chloroform-water interface. J. Solid State Chem. 2019, 278, 120912.

[3] Qin, Y.; Song, Y.; Huang, T.; Qi, L. Ionic liquid-assisted synthesis of thorned gold plates comprising three-branched nanotip arrays. Chem. Commun. 2011, 47, 2985-2987.

Round 2

Reviewer 2 Report

The manuscript is now suitable for publication.